# The Effect of Worktime Control on Overtime Employees’ Mental Health and Work-Family Conflict: The Mediating Role of Voluntary Overtime

**DOI:** 10.3390/ijerph19073767

**Published:** 2022-03-22

**Authors:** Jiaoyang Yu, Stavroula Leka

**Affiliations:** 1School of Education, Taiyuan Normal University, Jinzhong 030619, China; 2School of Medicine, University of Nottingham, Jubilee Campus, Nottingham NG8 1BB, UK; 3Business School, University College Cork, T12 K8AF Cork, Ireland; stavroula.leka@ucc.ie

**Keywords:** worktime control, voluntary overtime, mental health, work-family conflict

## Abstract

Overtime has become a widespread phenomenon in the current information age that creates a high speed working pace and fierce competition in the high technology global economy. Based on the time-regulation mechanism and effort-recovery model, we examined the effect of worktime control (WTC) on mental health and work-family conflict (WFC) among overtime employees, and whether voluntary overtime mediated the relationships. We also examined two separate dimensions of WTC (control over time-off and control over daily hours). The results showed that control over time-off was related to decreased depression, anxiety, stress and WFC, while control over daily hours was related to decreased stress and WFC. Generally, control over time-off was beneficial to females and employees with dependents. Furthermore, mediation results showed that voluntary overtime was a complete mediator of relationships between WTC and depression and anxiety as well as a partial mediator of the relationship between WTC and stress. However, this study did not find a mediating effect of voluntary overtime on the WTC-WFC relationship. Limitations and practical implications are discussed.

## 1. Introduction

Preceding studies have suggested that long working hours may increase the risk of various psychological problems, including stress [1], fatigue [2], psychological distress, anxiety and depression [3,4], as well as sleep disturbances and decreased cognitive functioning [5]. Recently, recovery from work has received more attention, and it is considered essential for maintaining health and work-life balance. According to the Effort-Recovery Model [6], incomplete recovery could be induced by extended working hours, which could be explained through two pathways: (i) time-based incomplete recovery, where employees may have less time to take a rest, enjoy family activities, and personal hobbies and exercise etc., and (ii) strain-based incomplete recovery, where the quality of recovery may be reduced because of the extension of strain produced in long working hours. Employees may feel too tired to make an effort with, or take part in, family matters. Accumulated incomplete recovery may further result in increased health problems [7]. This study investigated the influence of WTC on overtime employees’ mental health and WFC and further explored the mechanisms of whether WTC can facilitate voluntary overtime and consequently contribute to decreased negative effects caused by overtime work. This may help to fill the gap of limited and inconsistent findings in the literature of job resources and be beneficial to organizational management of overtime employees’ mental health.

### 1.1. Worktime Control (WTC)

WTC, an important psychosocial characteristic, may play a moderating effect on the relationships between overtime and mental health, work-family conflict, and job-related outcomes. This is particularly important taking into consideration the impact of the COVID-19 pandemic and new hybrid work arrangements. During the past decades, work-related flexibility in organizational practice has received greater attention [8]. Job control is related to the extent to which workers can decide how, when, and where to do their jobs. Job control could be considered as a key factor that can reduce the detrimental effect of overtime work on employees’ health and work-life interference and, in turn, optimize organizational outcomes [9]. ‘Temporal flexibility’ has become a popular type of flexibility, such as WTC. There are two types of work time flexibility, namely company-based and employee-oriented flexibility. The former refers to “the need of employers to extend, modify, or reduce work hours according to client or production needs” [10] (p. 503). Shift work and compulsory overtime are well-known examples. Employee-oriented work time flexibility refers to decision latitude regarding work time autonomy to adjust the work schedule to meet employee needs [11].

WTC, which is supposed to act as a specific type of job control, has been defined as “an employee’s possibilities of control over the duration, position, and distribution of work time” [10] (p. 503). Global WTC refers to the overall work time flexibility and schedule control. Additionally, there are multidimensional WTC forms. Well-known sub-dimensions include control over (1) Flextime: When to start and end the time of the workday, (2) The distribution of workdays over the work week, (3) Break control: When to take a break, (4) Leave control: When to take vacations or a day off, and (5) Overtime control: Whether and when to work overtime [12].

WTC has been indicated to ameliorate some of the detrimental effects of high job demands. It buffers the influences of long working hours on poor physical and mental symptom frequency [13], work-family interference [14], and sickness absence [15]. A systematic review by Nijp, Beckers, Geurts, Tucker and Kompier [12] also highlighted the importance of WTC for employees which can be identified as a predictor of health issues (i.e., burnout and sleep), work-life balance, and several job-related outcomes (i.e., job satisfaction, attitudes, and turnover).

### 1.2. WTC in Relation to Mental Health

Firstly, WTC is beneficial to health. Hino et al. [16] suggested that significantly higher psychological distress for long overtime compared to short overtime only happens in low job control situations, whereas there was no significant relationship between overtime hours and psychological distress among high job control employees. Depressive symptoms are significantly reduced within high control over daily working hours compared to low and moderate levels [17]. Żołnierczyk-Zreda et al. [18] found that employees working long hours but having high WTC reported a significantly lower level of mental health problems (e.g., somatic complaints and anxiety) than workers with low work time control. However, a lack of autonomy within determining the duration and position of work time may elevate the risk of sleep disturbances among employees [19]. Low work time control is significantly correlated with greater stress [20]. Through a stream of longitudinal cohort studies, Ala-Mursula and colleagues found that lack of WTC contributed to increased risks of health problems, while high work time autonomy decreased the adverse influence of overtime work on sickness absence and was beneficial in improving the balance of work and life [21].

### 1.3. WTC in Relation to Work-Family Conflict (WFC)

Increased WTC can help improve work-life balance. Work-family balance is positively influenced by increasing employees’ schedule control [21,22]. Similarly, Lingard et al. [23] reported that WTC was positively correlated with work-family enrichment. A randomized experiment, conducted by Angelici and Profeta [24] suggested that implementation of self-rostering facilitated employees’ work-life balance. Lee et al. [25] even claimed that the promotion of flexible work schedules particularly aimed to motivate employees’ work-life balance. Employees enjoy the benefits of work time autonomy to meet the demands of their work and personal lives. Therefore, organizations expecting to attract and retain talented employees could provide specific policies within a supportive work environment that enables employees to balance work and family [26]. Therefore, we hypothesized that WTC would be negatively related to depression, anxiety, stress and WFC.

**Hypothesis** **1a.**
*WTC will be negatively related to depression, anxiety and stress.*


**Hypothesis** **1b.**
*WTC will be negatively related to WFC.*


### 1.4. Gender and Dependents Status as Moderators between WTC and Outcomes

In terms of the influence of personal factors on mental health, previous studies have indicated that gender differences may contribute to differentiation in mental health outcomes, such as depression [27,28], anxiety [29,30,31] and stress [32,33]. A striking feature of prevalent mental health problems is that women are twice as likely as men to experience depression [34,35]. Men and women appear to show different responses in stress and also have different coping styles; in general terms, findings indicate that women seem to be more emotional and less rational than men [32]. This may be a factor underpinning the large body of preceding evidence showing that women complain more frequently of sleep disturbance than men [36]. WTC may be much more beneficial to employees who have dependents being taken care of. Greater family demands will have greater impact on WFC [37]. Worktime flexibility enables employees to balance work tasks and responsibility for dependents.

In addition, in terms of WFC, previous studies showed that women seemed to suffer increased work-family conflict compared to men [38]. However, a meta-analysis conducted by Byron [39] investigating the antecedents of WFC, indicated that demographic characteristics, such as gender, marital status and income, were related weakly to WFC and acted as poor predictors of WFC alone. We further hypothesized that gender and dependents status would be important personal variables that would affect the WTC—mental health and WFC associations.

**Hypothesis** **2a.**
*The interaction effects of control over time-off with gender and dependent status on mental health problems and WFC will be significant; high control over time-off will have greater influence on females and employees who have dependents.*


**Hypothesis** **2b.**
*The interaction effects of control over daily hours with gender and dependent status on mental health problems and WFC will be significant; high control over daily hours will have greater influence on females and employees who have dependents.*


### 1.5. Voluntary Overtime as the Mediator between WTC and Outcomes

Moreover, whether employees engage in involuntary overtime or voluntary overtime appeared to make a difference on the impact of overtime. The distinction between voluntary and involuntary overtime is assumed to be an important moderator between overtime work and well-being [8]. A small amount of previous research has identified some of the factors which may be related to the likelihood of choosing to do overtime even when it is voluntary, such as more job variety and proper compensation [40]. Caplan and Jones [41] stated that due to an inappropriate amount of work required to be completed in a given amount of time, overtime work occurred. Ishida et al. [42] claimed the group influence on workers that overtime work generated from implicit pressures from colleagues or supervisors. Moreover, the extrinsic and intrinsic motivation model was widely used in previous studies to illustrate work engagement at work [43,44].

However, there is still hardly any research investigating the influence of involuntary or voluntary overtime on employees’ mental health and WFC. One of the few exceptions, Watanabe and Yamauchi [45], was a quantitative study exploring the impact of involuntary and voluntary overtime on work-nonwork balance, finding that involuntary overtime has a negative influence on work-nonwork balance and voluntary overtime work has a positive effect. The mediation model hypothesized in this study may help identify and explain the mechanism or process that underlines an observed relationship between WTC and mental health as well as WFC. As the involuntary overtime and voluntary overtime dimensions were rarely investigated in previous research, and in the absence of previous research exploring the relationships between involuntary and voluntary overtime, mental health and WFC we propose the following hypotheses:

**Hypothesis** **3a.**
*Involuntary and voluntary overtime significantly affect mental health.*


**Hypothesis** **3b.**
*Involuntary and voluntary overtime significantly affect WFC.*


We hypothesized that WTC may enable employees to work overtime voluntarily thereby causing less mental health problems and WFC.

**Hypothesis** **4a.**
*Voluntary overtime mediates the effect of WTC on mental health.*


**Hypothesis** **4b.**
*Voluntary overtime mediates the effect of WTC on WFC.*


## 2. Materials and Methods

### 2.1. Participants and Recruitment Procedure

We recruited participants by liaising with organisations in the IT industry in China, which were distributed in different areas of China. They were China’s top 100 listed IT companies in the last three years, engaging in software and services, computer, communication, electronics manufacturing, internet and so on. A total of 25 organisations were invited by email and five organisations agreed to participate in this research. The percentage of the studied organizations in the whole population was 20%. Participants were employed by IT companies and worked full-time, whereas part-time employees were excluded. Participants were employees in any position in their company. The invitation email with a link to the online questionnaire was sent to all employees of these organisations by the department of human resources, inviting them to voluntarily participate in the study and informing them that the survey was anonymous and non-participation would not be identified or carry any negative consequences. At the beginning of the questionnaire survey, a consent form was provided to inform the participants what would be involved in the study, the anonymity and confidentiality of their answers, as well as to provide them with the opportunity to provide their consent to participate through the submission of their anonymous responses. One month after the invitation email was sent, 206 responses had been received. Therefore, a reminder email was sent and 59 more questionnaires were completed. A combined total of 265 valid responses to the online questionnaire were received.

### 2.2. Measures

#### 2.2.1. Demographics

Personal and job demographic data were collected, including gender, age, marital status, dependents’ situation, education level and income level. The investigation of these demographics was based on previous studies which suggested they are likely relevant to working overtime.

#### 2.2.2. Worktime Control (WTC)

The Worktime Control measure used in this study was the seven item scale from Ala-Mursula et al. [46]. This scale measures two aspects: control over daily hours and control over time-off, including flextime, schedule control, break control and leave control. Sample items include: “How much you can influence the starting and ending times of a workday?” and “How much you can influence the taking of breaks during the workday?” Responses to each item were given on a five-point scale (1 = very little, to 5 = very much). The measure had good internal consistency, with a Cronbach’s alpha of 0.82. The mean of each scale score was calculated and divided into tertiles to categorize low, moderate and high levels [47,48].

#### 2.2.3. Voluntary and Involuntary Overtime

Watanabe and Yamauchi’s [45] scale was used, which measures overtime control from four sub-dimensions, namely involuntary overtime work due to workload and conformity, voluntary overtime work due to extrinsic motivation and intrinsic motivation. The same structure was used in this study. However, the original scale could not be directly used in this study as their questionnaire was designed to measure nurses’ overtime control and some items were relevant to nurses’ specific work situations. Eight items were slightly modified to suit the current participants and the other seven items were kept from Watanabe and Yamauchi’s [45] scale. Cronbach’s α for workload, conformity, extrinsic motivation and intrinsic motivation in this study were 0.72, 0.91, 0.80 and 0.85, respectively. Table 1 lists modified items.

#### 2.2.4. Mental Health

The Chinese version of Depression Anxiety and Stress Scales (DASS-21) was used in this study, which was developed by researchers at the University of New South Wales (Austrilia). DASS-21 is an instrument with 21 items assessing current (“over the past week”) symptoms of depression, anxiety, and stress. Each of the three scales contains seven items. Each item is rated on a 4-point combined severity/frequency scale to measure the extent to which individuals have experienced each item over the past week. The scale ranges from 0 (did not apply to me at all) to 3 (applied to me very much, or most of the time). The sum of each scale is calculated, with higher scores representing more serious problems. The Chinese version of DASS-21 is adequately and appropriately translated and adapted, and has demonstrated cross-cultural reliability with Cronbach’s alphas: 0.83, 0.80, and 0.82 for the Depression, Anxiety, and Stress subscales, respectively, and 0.92 for the total DASS [49,50].

#### 2.2.5. Work-family Conflict (WFC)

The WFC measure was utilized in this study from Carlson et al. [51]. WFC is a bidirectional construct consisting of work interfering with family (WIF) and family interfering with work (FIW). Each direction contains three forms, namely time-based, strain-based and behaviour-based WFC. Responses were provided on a 5-point scale ranging from 1 (strongly disagree) to 5 (strongly agree), with higher scores representing more conflict. In the present study, WIF (three subscales with three items each) was measured (Cronbach’s α = 0.75), as this study focused on the impact of overtime work on employees’ families. Sample items were “The time I must devote to my job keeps me from participating equally in household responsibilities and activities” for time-based WIF, “I am often so emotionally drained when I get home from work that it prevents me from contributing to my family” for strain-based WIF and “Behaviour that is effective and necessary for me at work would be counterproductive at home” for behaviour-based WIF.

## 3. Results

Table 2 illustrates the basic demographic characteristics of 265 participants, including gender, age, marital status, the situation of dependents, educational level and income level.

We performed correlation analysis to examine associations between two separate dimensions of WTC (namely control over time-off and control over daily hours) and outcome variables. Table 3 presents detailed results of bivariate correlation analyses. Control over time-off was significantly negatively related to depression (*r* = −0.14, *p* = 0.022), anxiety (*r* = −0.18, *p* = 0.003), stress (*r* = −0.24, *p* < 0.001) and WFC (*r* = −0.38, *p* < 0.001). Control over daily hours was significantly negatively related to stress (*r* = −0.15, *p* = 0.015) and WFC (*r* = −0.28, *p* < 0.001). These results indicated support for Hypotheses 1a and 1b.

Hierarchical regression analyses were conducted to test the interaction effects of control over time-off and control over daily hours with gender and dependent status. Potential influential demographic characteristics variables were controlled and entered in step 1. Next, the categorical variable, control over time-off was entered in step 2; low control over time-off was the reference group. Finally, the interactions between control over time-off with gender and dependent status were entered in step 3. Table 4 indicates that after controlling for potential influential demographic characteristics, the interaction effects of control over time-off with gender and dependents were significant for depression, *R*^2^ = 0.130, Δ*R*^2^ = 0.046, *F*(4, 251) = 3.31, *p* = 0.012, 95% CI (6.32, 14.48); anxiety, *R*^2^ = 0.155, Δ*R*^2^ = 0.039, *F*(4, 251) = 2.88, *p* = 0.023, 95% CI (9.66, 17.25); stress, *R*^2^ = 0.180, Δ*R*^2^ = 0.044, *F*(4, 251) = 3.35, *p* = 0.011, 95% CI (9.01, 16.13) and WFC, *R*^2^ = 0.288, Δ*R*^2^ = 0.057, *F*(4, 251) = 5.06, *p* = 0.001, 95% CI (28.94, 42.01). Therefore, Hypothesis 2a was supported.

Compared to low control over time-off, high control over time-off had significantly greater influence on females than males in depression, *t*(251) = −2.89, *p* = 0.004 and compared to low control over time-off, moderate control over time-off had significantly greater influence on participants with dependents than participants without dependents in depression, *t*(251) = −2.04, *p* = 0.042.

Compared to low control over time-off, high control over time-off and moderate control over time-off had significantly greater influence on participants with dependents than participants without dependents in anxiety, *t*(251) = −3.20, *p* = 0.002 and *t*(251) = −2.07, *p* = 0.040, respectively; in stress, *t*(251) = −2.75, *p* = 0.006 and *t*(251) = −2.66, *p* = 0.008, respectively. However, the interaction effect of control over time-off with gender was not significant for anxiety and stress.

Compared to low control over time-off, high control over time-off had significantly greater influence on females than males in WFC, *t*(251) = −3.66, *p* < 0.001. The interaction effect of control over time-off with dependents status was not significant in WFC.

However, in terms of control over daily hours, Table 5 indicates that the interaction effects of control over daily hours with gender and dependents were not significant in depression, anxiety, stress and WFC. Therefore, Hypothesis 2b was not supported.

Correlation analysis was conducted to examine the correlations between involuntary overtime and voluntary overtime with depression, anxiety, stress and WFC. Table 6 shows that involuntary overtime was significantly positively correlated with higher depression, anxiety, stress and WFC. However, voluntary overtime was significantly negatively correlated with lower depression, anxiety, stress, and WFC. These findings indicated that Hypothesis 3 was supported.

Furthermore, hierarchical regression analyses were conducted to investigate the effect of involuntary overtime (including workload and conformity) and voluntary overtime (including extrinsic motivation and intrinsic motivation) on dependent variables after controlling for potential demographic characteristics. Demographic characteristics were entered in step one and four dimensions, namely workload, conformity, extrinsic motivation and intrinsic motivation were entered in step two.

In general, Hypotheses 3a and 3b were supported. Table 7 showed that all four independent variables appear to make different contributions to the explanation of dependent variables. Conformity and intrinsic motivation make significant contributions to depression (β = 0.50, *p* < 0.001 and β = −0.22, *p* = 0.002) and anxiety (β = 0.50, *p* < 0.001 and β = −0.19, *p* = 0.006); workload, conformity and intrinsic motivation make significant contributions to stress (β = 0.18, *p* = 0.003, β = 0.41, *p* < 0.001 and β = −0.17, *p* = 0.016, respectively); workload and conformity have significant effects on WFC (β = 0.27, *p* < 0.001 and β = 0.21, *p* = 0.001).

The mediation analysis showed the mediation effect of voluntary overtime on the relationships between WTC and mental health. The results (see Figure 1, Figure 2, Figure 3 and Figure 4) showed that WTC significantly influenced voluntary overtime, β = 0.43, *p* < 0.001. Voluntary overtime significantly influenced depression (β = −0.20, *p* < 0.001), anxiety (β = −0.12, *p* = 0.007), stress (β = −0.11, *p* = 0.006) and WFC (β = 0.33, *p* < 0.001). Additionally, there was a significant indirect effect of WTC on depression through voluntary overtime, β = −0.09, 95% CI [−0.15, −0.03], *P_M_* = 0.92; on anxiety, β = −0.05, 95% CI [−0.11, −0.004], *P_M_* = 0.49; on stress, β = −0.05, 95% CI [−0.11, −0.005], *P_M_* = 0.34. However, the indirect effect of WTC on WFC through voluntary overtime was not significant, β = −0.07, 95% CI [−0.17, 0.01]. Therefore, Hypothesis 4a was supported but Hypothesis 4b was not supported.

## 4. Discussion

The current study found that control over time-off and control over daily hours significantly affected mental health and WFC which is consistent with previous studies and adds further evidence to previous literature. Nijp, Beckers, Geurts, Tucker and Kompier’s [12] systematic review on the association between WTC and work-non-work balance, health and well-being suggested strong cross-sectional evidence for positive associations between WTC and work-non-work balance, whereas no consistent evidence was found for well-being. In terms of WFC, there are relatively consistent findings in previous research studies [52,53,54], whereas the evidence for mental health is inconsistent and limited. Regarding mental health, this study indicated that individuals with high or moderate control over time-off or daily hours reported less mental health problems (including depression, anxiety and stress) than those with low control over time-off or daily hours. The finding on stress is in line with previous studies that reported a significant negative association between WTC and stress [55,56]; in terms of the relationship of WTC with depression, it is consistent with previous studies [17,57], both of which indicated that albeit small effects of WTC on depressive symptoms were found, WTC plays a role in decreasing depressive symptoms. To the authors’ knowledge, there is limited research investigating the relationship between WTC, anxiety and depression. This research contributes towards filling the gap in this area and further studies are needed to confirm these findings.

Moreover, the impact is thought to be reversible, as long as the individual has sufficient autonomy on worktime, both leave-off and daily hours. However, if there is less control over worktime, then negative load effects may accumulate, resulting in increased mental ill health issues. Lack of WTC results in insufficient opportunity for recovery from work, and negative load effects may accumulate, resulting in prolonged fatigue and, eventually, poorer health [58]. In addition, this study found that control over time-off seemed to have higher influence on females than males in relation to their mental health and WFC, and control over daily hours in relation to WFC, which is in line with previous literature. The majority of researchers indicated that although the gender gap has been diminished over the last few decades, women still engage in more domestic work than men, and experience more stress in balancing working hours and time spent in unexpected family responsibilities [53,59]. Therefore, worktime flexibility would greatly benefit women. Moreover, this study found that control over time-off had significant impact on employees who had dependents than those who did not, which was rarely investigated specifically in the previous studies. Worktime flexibility has also been found to be related to higher organizational commitment and job satisfaction for those having family responsibilities [60].

Involuntary overtime was associated with increased mental health problems and WFC whereas voluntary overtime was associated with decreased mental health problems and WFC. These findings are in line with limited findings of previous studies. Watanabe and Yamauchi [61,62] demonstrated that the influence of overtime work differed by the reasons for working overtime, with involuntary overtime exerting an adverse impact on overtime employees’ mental health whereas voluntary overtime exerted the opposite effect. Golden and Wiens-Tuers [63] suggested that involuntary overtime was likely to impair well-being, induce work-family interference and low job satisfaction. Ota et al. [64] found that involuntary overtime work was associated with relatively high fatigue and low job satisfaction whereas voluntary overtime workers were non-fatigued and satisfied. In a similar vein, Van Der Hulst and Geurts [65] indicated that a high pressure to work overtime in combination with low rewards was associated with elevated risks of poor recovery, cynicism and negative work-home interference and in low reward situations, even a limited number of hours of involuntary overtime would induce adverse mental health problems.

Furthermore, based on Watanabe and Yamauchi [45], who identified two dimensions of involuntary overtime, namely workload and conformity; and two dimensions of voluntary overtime, namely extrinsic motivation and intrinsic motivation, another new finding of this study is the impact of these dimensions on mental health and WFC. Conformity is significantly negatively associated with employees’ mental health whereas intrinsic motivation is significantly positively associated with the same, which is consistent with previous findings. Guo et al. [66] stated that pressures could be not only from workload, but also from other persons, such as managers, which involuntary overtime work may generate from. Employees may suffer elevated ill mental health and fatigue when they work overtime in order to meet cultural or organizational customs or others’ expectations, whereas if employees are intrinsically motivated to engage in overtime work, they may suffer less mental health troubles and fatigue. Employees who have to engage in longer working hours beyond their intention are much more likely to experience fatigue than those who are internally willing and motivated to work long hours [2,62]. Additionally, undertaking overtime work due to workload puts employees in a much more stressful psychological and physical situation. Furthermore, involuntary overtime may increase WFC, consistent with previous literature [63,67].

The mediation effect of voluntary overtime in the relationship between WTC and mental health identified in this study seems to be a new finding, which explored the mechanism of how WTC plays a role in overtime employees. Time management as a source of control in line with the job demand-control (JDC) model could be used to illustrate this finding. Time management behaviour, a source of decision latitude, positively affects employees’ psychological health [68]. It may be that WTC gives employees more flexibility and autonomy on when and where to conduct overtime work and therefore it facilitates voluntary overtime that further causes less antipathy, cynicism and emotional exhaustion towards overtime and induces less adverse influences. High WTC enables employees to schedule their working hours and at the same time arrange their overtime work according to their current and actual situation, which arouses the impetus for the work and collision avoidance. As the overtime work is planned by employees themselves, they are more likely to engage in voluntary overtime work. The mediation model identified in this study highlights the effect of worktime flexibility on facilitating employees’ voluntary overtime further decreasing mental health problems caused by overtime work, which is a theoretical contribution for this area.

### 4.1. Limitations

The cross-sectional design applied in this research does not allow us to ascertain causal inferences about the associations found. Cross-sectional studies generally collect routine data to answer the specific question, whereas confounding factors or other variables that affect the relationship between the putative cause and effect are not included. Cross-sectional studies are generally carried out at one time and give no indication of the sequence of events [69]. We asked people if they had overtime in the previous six months, but we did not have information about their subsequent and earlier overtime information. Therefore, it is normally not possible to establish a cause and effect relationship. Associations identified may be tentative and difficult to interpret. Further studies need to be developed that explore the relationships in greater depth. Furthermore, we did not explore the job status/seniority of the respondents which might have had an impact on their decision to work overtime. This should be further explored in future studies.

### 4.2. Implications

The mediation models found in this research have further sought to identify and explain the mechanism or process that underlines an observed relationship between WTC and mental health, and WFC via the mediating variable, voluntary overtime. The conceptual model about voluntary and involuntary overtime put forward in this research is a new finding and a contribution to theory and future research in this area. Furthermore, improving worktime flexibility may particularly benefit overtime employees significantly who had frequent or long overtime working hours in terms of reducing their mental health problems and WFC. The results of the study show that gender matters in the outcomes of worktime flexibility; that control over time-off may be more beneficial to female and overtime employees who have dependents to take care of. This study suggests that when the organization considers undertaking the practice of providing employees with worktime autonomy job resources at work, gender and personal situations should be taken into account in order to maximize the function of worktime autonomy.

Besides WTC interventions, facilitating employees to engage in voluntary overtime and preventing compulsory overtime working is likely another way to reduce negative outcomes caused by overtime. Even though Olds and Clarke [70] found that the risks of adverse events still increased if nurses engaged in voluntary overtime more than 4 h a week on average and Watanabe and Yamauchi [45] indicated that overtime work may ultimately induce adverse impacts on nurse well-being, voluntary overtime was also recommended as another moderator likely to induce less adverse emotions and impacts than compulsory overtime. In particular, the present research indicated that influences of overtime on employees’ mental health and WFC depend on the level of voluntary overtime.

Therefore, this research will valuably inform the organization to carefully assess the reasons for overtime and explore potential ways to facilitate voluntary overtime and avoid involuntary overtime for the promotion of occupational health and performance. This recommendation is particularly crucial regarding long overtime working employees, as WTC may function as an important work characteristic for reducing adverse impacts of long overtime. Relevant interventions could be tailored depending on the organization’s level and nature of overtime work and individual employees’ overtime situation. Appropriate interventions could be taken into account based on the study findings such as the interaction effect of control over time-off with gender by human resource managers to improve employee well-being and performance. We conclude with some directions aimed at improving overtime employees’ mental health, work-life balance and boosting job-related outcomes and recommendations for future research.

## 5. Conclusions

This paper presents an exploration of the relationship of WTC, voluntary or involuntary overtime and employees’ mental health and WFC. This study found one of the dimensions of WTC, control over time-off, was related to decreased depression, anxiety, stress and WFC; the other dimension, control over daily hours was related to decreased stress and WFC. Compared to control over daily hours, control over time-off exerted more effect on females and employees who had dependents to take care of. In addition, WTC may facilitate voluntary overtime which is associated with less adverse outcomes than involuntary overtime. Employees who engage in higher voluntary overtime work are more likely to experience less mental health problems and WFC. Further research is needed to explore the process or mechanism in WTC—mental health and WFC relationships via voluntary overtime.

Due to the limitation of cross-sectional studies, there is a clear need for well-designed longitudinal and interventional studies to delineate the impact of flexible work time conditions on employees’ wellbeing and WFC and how to facilitate voluntary overtime in employees of different job status. Future research can investigate each sub-dimension of WTC in relation to its respective effect on overtime’s outcomes. Furthermore, more research is needed to demonstrate whether facilitating voluntary overtime decreases adverse influences and investigate how culture context affects employees’ behaviour engaged in overtime; for instance, how the employer or manager might constraint the employee’s overtime decision.

## Figures and Tables

**Figure 1 ijerph-19-03767-f001:**
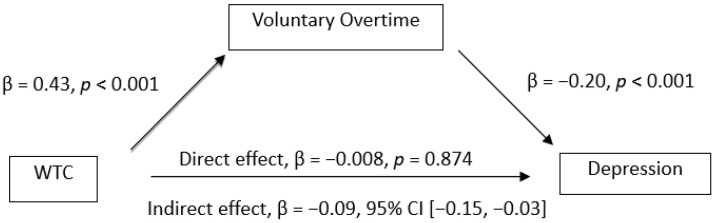
Observed relationships between WTC, voluntary Overtime and depression.

**Figure 2 ijerph-19-03767-f002:**
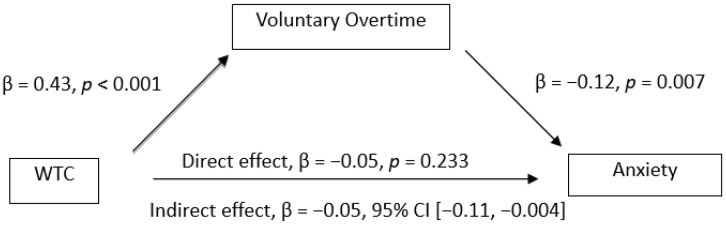
Observed relationships between WTC, voluntary overtime and anxiety.

**Figure 3 ijerph-19-03767-f003:**
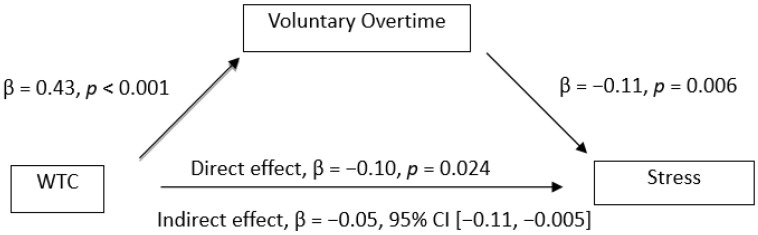
Observed relationships between WTC, voluntary overtime and stress.

**Figure 4 ijerph-19-03767-f004:**
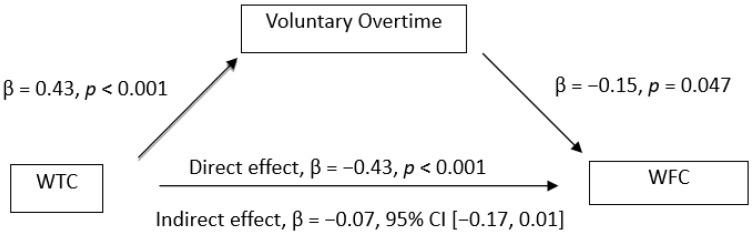
Observed relationships between WTC, voluntary overtime and WFC.

**Table 1 ijerph-19-03767-t001:** Modified items of original voluntary and involuntary overtime scale.

Original Items	Modified Items
I feel uneasy to go back when my boss and superiors still remain in my ward.	I feel uneasy to go back when my boss and superiors still remain in my office.
It is unfavorable to go back when others still remain in my ward.	It is unfavorable to go back when others still remain in my office.
I have to work overtime because many of my colleagues in my ward are engaged in overwork.	I have to work overtime because many of my colleagues in my office are engaged in overwork.
I have to work overtime because there is so much work to be done in my ward.	I have to work overtime because there is so much work to be done.
The work hours in my ward tends to be long because there are sudden deterioration of the patients and emergency admissions.	I have to work overtime because there are interruptions or unexpected and emergent tasks during the workday.
I have to work overtime because the manpower in my ward is in shortage.	I have to work overtime because the manpower in my office is in shortage.
I have to work overtime because I have to write the nurse records.	I have to work overtime because some tasks need to be completed in a limited time.
I cannot deal with the degree of severity and the number of assigned patients during the fixed time.	I have to work overtime because some job tasks are difficult and complicated.

**Table 2 ijerph-19-03767-t002:** Demographic characteristics of participants.

Variables	*n*	%
Gender		
Male	144	54.3
Female	121	45.7
Age		
21–30	122	46.0
31–40	109	41.1
41–50	33	12.5
51–60	1	0.4
Marital status		
Single	89	33.6
Married	157	59.2
Cohabiting	14	5.3
Divorced	4	1.5
Widowed	1	0.4
Dependents status		
Yes	151	57.0
No	114	43.0
Educational attainments		
Bachelor	168	63.4
Master	97	36.6
Income level (RMB)		
Below 3000	38	14.3
3000–5000	88	33.2
5001–7000	49	18.5
7001–9000	59	22.3
Above 9000	31	11.7

**Table 3 ijerph-19-03767-t003:** Correlation analysis (*n* = 265).

	1	2	3	4	5	6
1. Control over time-off						
2. Control over daily hours	0.61 ***					
3. Depression	−0.14 *	−0.11				
4. Anxiety	−0.18 **	−0.10	0.83 ***			
5. Stress	−0.24 ***	−0.15 *	0.79 ***	0.82 ***		
6. WFC	−0.38 ***	−0.28 ***	0.48 ***	0.52 ***	0.62 ***	0.57 ***
*M*	110.97	80.49	80.83	80.87	90.31	270.23
*SD*	40.01	30.14	40.79	40.51	40.30	80.46
α	0.85	0.81	0.91	0.88	0.87	0.95

*Note*: * *p* < 0.05. ** *p* < 0.01. *** *p* < 0.001.

**Table 4 ijerph-19-03767-t004:** Hierarchical multiple regression analysis explaining mental health variables and WFC from control over time-off.

Predictor	Depression		Anxiety		Stress		WFC	
	Δ*R*^2^	β	Δ*R*^2^	β	Δ *R*^2^	β	Δ*R*^2^	β
Step 1	0.061 *		0.075 **		0.087 **		0.103 ***	
Control variables ^a^								
Step 2	0.023 *		0.041 **		0.049 **		0.128 ***	
MCT vs. LCT		−0.17 *		−0.20 **		−0.17 *		−0.32 ***
HCT vs. LCT		−0.16 *		−0.23 **		−0.27 ***		−0.42 ***
Step 3	0.046 *		0.039 *		0.044 *		0.057 **	
MCT × G		−0.23		−0.17		−0.12		−0.01
HCT × G		−0.31 **		−0.08		−0.20		−0.36 ***
MCT × D		−0.26 *		−0.40 **		−0.34 **		−0.03
HCT × D		−0.02		−0.24 *		−0.30		−0.13
Total *R*^2^	0.130 **		0.155 ***		0.180 ***		0.288 ***	

*Note.* CI = confidence interval. HCT = high control over time-off. MCT = moderate control over time-off. LCT = low control over time-off. G = gender. D = dependents status. ^a^ Control variables included gender, age, marital status, dependents status, education level and income. * *p* < 0.05. ** *p* < 0.01. *** *p* < 0.001.

**Table 5 ijerph-19-03767-t005:** Hierarchical multiple regression analysis explaining mental health variables and WFC from control over daily hours.

Predictor	Depression		Anxiety		Stress		WFC	
	Δ*R*^2^	β	Δ*R*^2^	β	Δ*R*^2^	β	Δ*R*^2^	β
Step 1	0.061 *		0.075		0.087 **		0.103 ***	
Control variables ^a^								
Step 2	0.029 *		0.027 *		0.045 **		0.103 ***	
MCD vs. LCD		−0.20 **		−0.19 **		−0.23 **		−0.31 ***
HCD vs. LCD		−0.13		−0.13		−0.19 **		−0.35 ***
Step 3	0.013		0.015		0.016		0.022	
MCD × G		−0.02		−0.004		−0.11		−0.16
HCD × G		−0.16		−0.06		−0.17		−0.23 *
MCD × D		−0.02		−0.17		−0.04		−0.06
HCD × D		−0.11		−0.21		−0.15		−0.12
Total *R*^2^	0.104 **		0.117 **		0.147 ***		0.228	

*Note.* CI = confidence interval. HCD = high control over daily hours. MCD = moderate control over daily hours. LCD = low control over daily hours. G = gender. D = dependent status. ^a^ Control variables included gender, age, marital status, dependent status, education level and income. * *p* < 0.05. ** *p* < 0.01. *** *p* < 0.001.

**Table 6 ijerph-19-03767-t006:** Involuntary and voluntary overtime and outcome variables correlations.

	Involuntary Overtime	Voluntary Overtime
Depression	0.53 ***	−0.29 ***
Anxiety	0.52 ***	−0.21 ***
Stress	0.59 ***	−0.24 ***
WFC	0.51 ***	−0.25 ***

*** *p* < 0.001.

**Table 7 ijerph-19-03767-t007:** Hierarchical multiple regression analysis explaining outcome variables from involuntary and voluntary overtime.

Predictor	Depression		Anxiety		Stress		WFC	
	Δ*R*^2^	β	Δ*R*^2^	β	Δ*R*^2^	β	Δ*R*^2^	β
Step 1	0.061 *		0.091 **		0.087 **		0.103 ***	
Control variables ^a^								
Step 2	0.335 ***		0.321 ***		0.296 ***		0.201 ***	
Workload		0.03		0.06		0.18 **		0.27 ***
Conformity		0.50 ***		0.50 ***		0.41 ***		0.21 **
Extrinsic motivation		0.03		0.05		0.05		−0.06
Intrinsic motivation		−0.22 **		−0.19 **		−0.17 *		−0.10
Total *R*^2^	0.335 ***		0.413 ***		0.383 ***		0.304 ***	

*Note.* CI = confidence interval. ^a^ Control variables include gender, age, marital status, dependent status, education level and income. * *p* < 0.05. ** *p* < 0.01. *** *p* < 0.001.

## Data Availability

The data that support the findings of this study are available from the corresponding author upon reasonable request.

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
