# Peer review of "The Effect of Worktime Control on Overtime Employees’ Mental Health and Work-Family Conflict: The Mediating Role of Voluntary Overtime"

_ijerph, 2022, doi:10.3390/ijerph19073767_

Round 1

Reviewer 1 Report

From my point of view, the article presents a high scientific and merit level. I suggest adding information on further research in the topic.

Author Response

Point 1: From my point of view, the article presents a high scientific and merit level. I suggest adding information on further research in the topic.

Response 1: We thank the reviewer for their valuable comments. We have added more information in the paper. For instance, in the conclusion section, we added “Due to the limitation of cross-sectional studies, there is a clear need for well-designed longitudinal and intervention studies to delineate the impact of flexible work time conditions on employees’ wellbeing and WFC and how to facilitate voluntary overtime in employees of different job status. Future research can investigate each sub-dimension of WTC in relation to its respective effect on overtime’s outcomes. Furthermore, more research is needed to demonstrate whether facilitating voluntary overtime decreases adverse influences and investigate how culture context affects employees’ behaviour engaged in overtime; for instance, how the employer or manager might constraint the employee’s overtime decision.”

Reviewer 2 Report

Thank you for requesting my review of this manuscript. I have read it carefully and can say that a good job has been done.

The Introduction section and the document in general is clearly written and quoted, providing rigorous information.

The results provided are of value and interest to the area of study and are also consistent with the theoretical framework and objective set.

The methodology used is a well-planned research design that provides reasonable and consistent data and results. Therefore, this section is appropriate for the object of study and has a consistent and rigorous design.

The contributions made throughout the document are well documented and justified, backed by previous studies that justify each statement, so the manuscript appears to be a high quality research.

However, in order to further improve the quality of the study, I propose that the authors take the following actions:

  • Add a Practical Applications section.
  • Explain in more detail the conclusions reached and add further previous literature to confirm or refute the conclusions drawn.

Author Response

Point 1: Thank you for requesting my review of this manuscript. I have read it carefully and can say that a good job has been done.

The Introduction section and the document in general is clearly written and quoted, providing rigorous information.

The results provided are of value and interest to the area of study and are also consistent with the theoretical framework and objective set.

The methodology used is a well-planned research design that provides reasonable and consistent data and results. Therefore, this section is appropriate for the object of study and has a consistent and rigorous design.

The contributions made throughout the document are well documented and justified, backed by previous studies that justify each statement, so the manuscript appears to be a high quality research.

However, in order to further improve the quality of the study, I propose that the authors take the following actions:

Add a Practical Applications section.

Response 1: Thank you for your valuable comments. We have added more information in terms of the study’s application which can be found at the end part of the discussion section.

Point 2: Explain in more detail the conclusions reached and add further previous literature to confirm or refute the conclusions drawn.

Response 2: We have provided and added the relevant information in the discussion section.

Reviewer 3 Report

In reviewing the study, I have taken into account three fundamental aspects, considering that they have used a cross-sectional observational design.
- Are the results of the study valid?
In this regard, it should be noted that the topic addressed is clearly defined; the study analyzes the effects of time management on mental health and work-family conflicts, in a scenario in which overtime is necessary for competitive reasons.
However, the procedure for recruiting and selecting the workers does not facilitate the external validity of the results. We cannot say with certainty that the results of the study are extensible to other situations for the following reasons: No information is provided on the companies in which the people who participated in the study were recruited, except that they operate in the Chinese technology industry sector, but they are many and very diverse, and their level of competitiveness is probably different. To solve this problem, the reader should be offered at least three indicators of competitiveness: Economic Growth, Inflation and Exchange Rate and Trade Balance.
Certainly, on this problem, we could think that knowing these indicators for the whole of the Chinese High-Tech industry would be enough, regardless of the fact that they are not similar for each of the selected companies. 
It would be very difficult to accept that workers with different positions in their companies are comparable in terms of time management, especially when I start from an effort-reward model. In general, the greater the job responsibility, the greater the perception of time management. Therefore, I believe that job status is a variable that must be taken into account in the regression models carried out by the authors.
- What are the results?
The results highlight a truism, time control has a differential effect on mental health and work-family conflict depending on whether overtime work is done voluntarily or not; i.e., "If you love scabies, they don't hurt".
- Can they be applied to other jobs and work contexts?
From what has been expressed in the previous paragraphs, it can be deduced that the main problem of the study is the generalization of the results, since such an important variable in the regulation of working time, such as work status, has not been considered.

Author Response

Point 1: In reviewing the study, I have taken into account three fundamental aspects, considering that they have used a cross-sectional observational design.
- Are the results of the study valid?
In this regard, it should be noted that the topic addressed is clearly defined; the study analyzes the effects of time management on mental health and work-family conflicts, in a scenario in which overtime is necessary for competitive reasons.
However, the procedure for recruiting and selecting the workers does not facilitate the external validity of the results. We cannot say with certainty that the results of the study are extensible to other situations for the following reasons: No information is provided on the companies in which the people who participated in the study were recruited, except that they operate in the Chinese technology industry sector, but they are many and very diverse, and their level of competitiveness is probably different. To solve this problem, the reader should be offered at least three indicators of competitiveness: Economic Growth, Inflation and Exchange Rate and Trade Balance.
Certainly, on this problem, we could think that knowing these indicators for the whole of the Chinese High-Tech industry would be enough, regardless of the fact that they are not similar for each of the selected companies. 
It would be very difficult to accept that workers with different positions in their companies are comparable in terms of time management, especially when I start from an effort-reward model. In general, the greater the job responsibility, the greater the perception of time management. Therefore, I believe that job status is a variable that must be taken into account in the regression models carried out by the authors.

Response 1: We thank the reviewer for their useful comments. The information about the procedure for recruiting and selecting the workers and recruited organisations was further elaborated in the section Participants and Recruitment Procedure. The demographic characteristics were controlled in the regression models. This study involved full-time employees in recruited organisations but it did not distinguish the employees’ job status. Therefore, we are unable to compare among employees with different job status in terms of the worktime flexibility. We have added this as a limitation to the study and suggestion for further research.

Point 2: What are the results?
The results highlight a truism, time control has a differential effect on mental health and work-family conflict depending on whether overtime work is done voluntarily or not; i.e., "If you love scabies, they don't hurt".

Response 2: We investigated the impacts of separate dimensions of worktime control, namely control over time-off and control over daily hours on overtime employees, taking gender and dependents status into consideration. Compared to control over daily hours, control over time-off exerted more effect on females and employees who had dependents to take care of. Therefore, this study suggests that when the organization considers undertaking the practice of providing employees with worktime autonomy job resources at work, gender and personal situations should be taken into account in order to maximize the function of worktime autonomy. In addition, the mediation models in this research sought to identify and explain the mechanism or process that underlines an observed relationship between WTC and mental health, and WFC via the mediating variable, voluntary overtime. They indicated that besides WTC interventions, facilitating employees to engage in voluntary overtime and preventing compulsory overtime working is another way to reduce negative outcomes caused by overtime.

Point 3: Can they be applied to other jobs and work contexts?
From what has been expressed in the previous paragraphs, it can be deduced that the main problem of the study is the generalization of the results, since such an important variable in the regulation of working time, such as work status, has not been considered.

Response 3: The generalisation of results is not possible as the study is cross-sectional which we have highlighted in the limitations of the study. Once again, we did not focus on job status specifically, which has been identified as a limitation of the study and suggestion for future research.

Reviewer 4 Report

Dear Authors,

Your paper is very interesting. However, and although its interest, some improvements are needed, namely:

What is the percentage of the studied organizations in the whole population?
How did authors select these organizations?
In what way are the obtained responses associated with each of the 5 organizations that accepted to participate in the study?
These issues need to be clarified.
In the Conclusion section, authors should highlight the main theoretical and practical contributions of this study to this research field.

Author Response

Point 1: Your paper is very interesting. However, and although its interest, some improvements are needed, namely:

What is the percentage of the studied organizations in the whole population? How did authors select these organisations?

Response 1: We thank the reviewer for their comments. This information has been added in section 2.1. Participants and Recruitment Procedure.

Point 2: In what way are the obtained responses associated with each of the 5 organizations that accepted to participate in the study?

Response 2: Our analysis does not focus on differences between the 5 organisations. The main interest of the study is on the specific sector.

Point 3: In the Conclusion section, authors should highlight the main theoretical and practical contributions of this study to this research field.

Response 3: The information about theoretical and practical contributions of this study is presented at the end of the discussion section.

Round 2

Reviewer 3 Report

I thank you for the attention you have given to my comments. In my opinion, the writing is more understandable now, as the reader will have more complete information on the characteristics of the companies in which the sample was recruited, and by highlighting the limitations of the study can also get a more accurate idea of the scope of the results. On the other hand, although the design of the study can be improved in future works (as you know, this type of design provides very weak evidence), the fact is that the possible shortcomings of the study are solved with the complete statistical analysis that is carried out. Therefore, I can only congratulate you.